# Nanoscale Materials Applying for the Detection of Mycotoxins in Foods

**DOI:** 10.3390/foods12183448

**Published:** 2023-09-15

**Authors:** Xiaochun Hu, Huilin Li, Jingying Yang, Xintao Wen, Shuo Wang, Mingfei Pan

**Affiliations:** 1Key Laboratory of Food Quality and Health of Tianjin, Tianjin University of Science and Technology, Tianjin 300457, China; hxchhxxzdq@163.com (X.H.); LHLtust@163.com (H.L.); yangjy0823@126.com (J.Y.); w2568665715@163.com (X.W.); s.wang@tust.edu.cn (S.W.); 2State Key Laboratory of Food Nutrition and Safety, Tianjin University of Science and Technology, Tianjin 300457, China

**Keywords:** mycotoxins, nanoscale materials, accurate and rapid detection, food

## Abstract

Trace amounts of mycotoxins in food matrices have caused a very serious problem of food safety and have attracted widespread attention. Developing accurate, sensitive, rapid mycotoxin detection and control strategies adapted to the complex matrices of food is crucial for in safeguarding public health. With the continuous development of nanotechnology and materials science, various nanoscale materials have been developed for the purification of complex food matrices or for providing response signals to achieve the accurate and rapid detection of various mycotoxins in food products. This article reviews and summarizes recent research (from 2018 to 2023) on new strategies and methods for the accurate or rapid detection of mold toxins in food samples using nanoscale materials. It places particular emphasis on outlining the characteristics of various nanoscale or nanostructural materials and their roles in the process of detecting mycotoxins. The aim of this paper is to promote the in-depth research and application of various nanoscale or structured materials and to provide guidance and reference for the development of strategies for the detection and control of mycotoxin contamination in complex matrices of food.

## 1. Introduction

To date, food safety remains one of the major issues of widespread concern worldwide. The presence of toxic and hazardous substances in food is an important aspect that contributes to food safety problems [1,2]. Foods such as grains, oils, and fats are prone to contamination by fungi such as *Aspergillus*, *Penicillium*, and *Fusarium* at various stages, including production, processing, storage, and transportation [3,4,5]. Under conditions of high temperature and humidity, these microorganisms can produce and accumulate mycotoxins and secondary metabolites that serve as typical food contaminants. Mycotoxins are highly toxic, widely contaminated, and difficult to remove [6,7,8]. They can enter and enrich the food chain through foodstuffs and animal-derived products such as meat, milk, and eggs and ultimately penetrate into organisms, causing reproductive abnormalities, immunosuppression, cancer, and other serious diseases, which pose a serious threat to human health [9,10]. In addition, most fungi are capable of producing multiple toxins simultaneously, making the co-contamination of food with multiple toxins highly common. The cumulative or synergistic effects of these toxins can lead to more significant toxic effects than single toxins [11,12], further highlighting the importance of controlling and monitoring mycotoxins in food. Consequently, the World Health Organization (WHO), the European Food Safety Authority (EFSA), the Food and Agriculture Organization of the United Nations (FAO), and the Codex Alimentaria Commission (Codex Alimentaria) have jointly established limits and detection requirements for biotoxins, including mycotoxins [13,14] (Table 1). It is essential to strengthen the research on specific, sensitive, rapid, and reliable strategies for mycotoxins detection in food to safeguard human health effectively [15,16].

The development of nanotechnology and materials science has led to the research of various new nanoscale materials, and different carbon-based, metal-based, and semiconductor-based nanoscale materials have been developed and applied in various fields [17,18]. Notably, these nanoscale materials exhibit extraordinary properties that offer innovative solutions for the precise and expeditious detection of trace contaminants, such as mycotoxins, in food [19,20,21]. In recent years, researchers have developed many efficient and accurate detection strategies based on nanoscale materials with various structures and properties. These innovations have assumed a pivotal role in the monitoring and identification of mycotoxins within food products [22,23]. The diverse structural and functional characteristics of nanoscale materials result in significant variations in their roles within specific applications. This inherent diversity often leads to ambiguity and uncertainty in the preparation and utilization of nanoscale materials. Additionally, there is a limited in-depth exploration of the mechanisms for enhancing sensitivity, accuracy, and other characteristics in mycotoxin detection strategies based on these materials. This study summarizes and compares recent studies on the accurate and rapid detection of mycotoxins in food based on nanoscale materials, with an emphasis on the role played by nanoscale materials with different properties in the detection process. This review provides valuable guidance for food safety monitoring and the development of advanced mycotoxin detection strategies.

## 2. Nanoscale Materials for Instrumental Analysis of Mycotoxins 

Currently, instrumental analysis techniques based on chromatographic separation, mass spectrometry, or spectroscopy remain the primary strategies for accurately detecting mycotoxins in food, widely accepted as standardized methods by international organizations [24,25,26]. Large-scale analytical instruments, typically equipped with sensitive detectors and data analysis modules, can successfully detect trace levels of toxin targets with advantages of accuracy, reproducibility, and reliability [27,28]. However, various mycotoxins may coexist at extremely low concentrations in food, and considering the complexity of food matrices, it is necessary to purify the food matrix during the detection process while achieving the enrichment of low-concentration mycotoxins to meet the requirements of instrument analysis [29]. In response to this challenge, novel purification materials with nanoscale features or exceptional structural characteristics have been continuously developed and used in combination with various large-scale analytical instruments, such as chromatography and mass spectrometry, achieving accurate and sensitive detection of mycotoxins in complex food matrices [30,31,32]. Table 2 illustrates the application of various nanoscale materials in solid-phase extraction (SPE) and solid-phase microextraction (SPME) processes for the detection of mycotoxins in food.

### 2.1. Absorbent for SPE

The SPE process is the most commonly used pretreatment method for complex food matrices, which can purify the matrix and enrich trace substances at the same time. This process requires only a small amount of organic solvent and has good reproducibility [48,49,50]. The property of SPE sorbents determines the effectiveness of SPE as a preprocessing technique [51]. Various nanoscale or microscale materials typically possess a large surface area, enabling the loading of numerous specific recognition groups and achieving specific recognition of trace mycotoxins in complex matrices [52,53]. This requirement is essential for excellent SPE purification materials.

Nano-silica (SiO_2_) is easily prepared and possesses a large pore volume and specific surface area. It exhibits excellent hydrophilicity and can be easily surface-modified and combined with other materials [54,55]. As a result, it is widely used as SPE sorbent for the purification of food matrices. Yuan et al. employed humic-acid-bonded silica (HAS) material for the SPE purification of lipid matrices, followed by high-performance liquid chromatography and photochemical post-column reactor fluorescence spectrum (HPLC-PHRED-FLD) to simultaneously quantify aflatoxin (AF) and benzo(a)pyrene (BaP) and evaluated the extraction effectiveness and efficiency [39]. The HAS adsorbent has outstanding adsorption properties due to the large number of functional group hydrogen bonding, hydrophobicity, and π-π interactions, which minimize the pretreatment time and the amounts of organic solvents. It can efficiently and stably adsorb two targets from the lipid matrix and obtain accurate detection results (limits of quantification (LOQs), 0.05–0.30 µg kg^−1^; limits of detection (LODs), 0.01–0.09 µg kg^−1^). Compared with a single type of SPE material, the composite SPE material composed of multiple nanoscale materials can combine the advantages of various materials in a targeted manner. This not only helps to improve the purification efficiency but also significantly improves the selectivity of the target compound. Especially in high-throughput, multi-target mycotoxin detection, composite SPE materials have obvious advantages. Wang and his team compared the performance of composite SPE materials composed of different types and dosages of multi-walled carbon nanotubes (MWCNTs) and five different typical adsorbents (i.e., octadecylsilyl (C_18_), hydrophilic–lipophilic balance (HLB), mixed-mode cationic exchange (MCX), silica gel, and amino-propyl (NH_2_)) in purifying corn and wheat matrices and extracted a total of 21 mycotoxins [41]. The combination of MWCNTs (20 mg) and C_18_ (200 mg) was demonstrated to be the most effective, significantly reducing the matrix effect, enabling the high-throughput screening of various mycotoxins, and greatly improving the detection efficiency. The study of Han et al. combined carbon-based nanomaterial graphene (rGO) with stable chemical properties, a high specific surface area, and a strong adsorption capacity with gold nanoparticles (AuNPs), which effectively overcame its irreversible aggregation problem in solution [36]. Compared with commercial SPE materials, their novel nanoadsorbent rGO/AuNPs showed comparable or even better adsorption and purification effects at a lower cost. A satisfactory linear quantification range (0.02–0.18 ng mL^−1^, R^2^ ≥ 0.992) was obtained for nine mycotoxins in milk in combination with ultra-performance liquid chromatography–tandem mass spectrometry (UPLC-MS/MS) analysis, which laid the foundation for the further development of an effective method for high-throughput and rapid screening of multi-mycotoxins. Although such nanoscale composite SPE materials have largely enhanced the purification and detection rates, they are not specific enough to accurately purify or enrich for a single mycotoxin. Therefore, the development of nanoscale pretreatment materials that are both efficient and specifically recognized for the extraction of mycotoxins from food products is of great importance.

Molecularly imprinted polymers (MIPs) are a class of chemically synthesized materials with the specific recognition capability to target molecules, which are called “artificial antibodies (Abs)” [56,57,58]. Due to the remarkable stability and selectivity, MIPs have been widely used as sorbents for the extraction of various chemical substances [59,60,61]. Dalibor’s team compared the prepared MIPs with selective recognition and binding sites for zearalenone (ZEN) with the non-selective reversed-phase C_18_ extractant and evaluated the difference in the analytical characteristics of the two extractants during the extraction process [62]. Due to the use of online SPE, the two detection strategies based on the high-performance liquid chromatography (HPLC) of MIPs or C_18_ absorbents established in this study overcame the drawbacks of time-consuming and manual sample pretreatment in ZEN detection. Unfortunately, the two SPE detection strategies were similar in terms of linear range, sensitivity, reproducibility, and even no significant difference in specificity identification, which was inferred to be caused by the strong affinity of the esterophilic target ZEN on the C_18_ sorbent. Metal–organic frameworks (MOFs) are a class of crystalline materials formed by the coordination of metal ions or clusters with organic ligands [63,64]. They are characterized by a high specific surface area, large porosity, ease of synthesis, thermal stability, and tenability [65,66]. Liang’s team attached MIPs to the surface of MOF material UiO-66-NH_2_ via the precipitation aggregation method as an adsorbent for SPE, which was used for the adsorption and quantification of aflatoxins (AFB_1_, AFB_2_, AFG_1_, and AFG_2_) in grains, and the adsorption capacity was compared with that of commercial SPE [40]. In this study, the surface of UIO-66-NH_2_ was modified by grafting glycidyl methacrylate (GMA), which effectively preserved the interaction between the monomer and the virtual template and formed hydrogen bonding sites. The prepared novel surface-imprinted polymer materials were uniform and stable, and the unique pore structure could effectively improve the selective adsorption capacity of polymer materials. Secondly, due to the large specific surface area of MOFs and the high specificity of MIPs, it shows excellent affinity and selectivity for aflatoxins, and it is a rapid, cheap, efficient, and reusable method. Unfortunately, although the problem of high cost and high toxicity of the target as a template has been solved, it still fails to selectively adsorb a single target substance.

Magnetic SPE is a new SPE method that has attracted extensive attention in the field of separation science because of its convenient, rapid, and efficient adsorption separation in a magnetic field [67,68]. Magnetically functionalized nanomaterials such as metal oxides, polymers, and organic frameworks are used in enrichment and separation processes for different targets [69,70,71]. Covalent organic frameworks (COFs) are a new type of crystalline material with the advantages of a large specific surface area, high porosity, abundant functional groups, and good thermal and chemical stability [72,73]. They can be combined with magnetic nanoparticles to increase pretreatment extraction materials’ porosity and specific surface area. In the study of Nie et al. [34], a magnetic COF nanomaterial Fe_3_O_4_@COF (TAPT-DHTA) was prepared via a simple template precipitation polymerization method, which was applied to simultaneously enrich nine mycotoxins in fruits. Combined with the analysis of ultrahigh-performance liquid chromatography in combination with tandem mass spectrometry (UHPLC-MS/MS), a wide linear range (0.05–200 μg kg^−1^) and a low LOD (0.01–0.5 μg kg^−1^) for nine targeted mycotoxins were achieved. Notably, the Fe_3_O_4_@COF adsorbent prepared in the study was rich in aromatic rings and carbonyl groups and, thus, can effectively enrich the target toxins through strong π-π interactions and hydrogen bonding. Zhang et al. [74] designed an effective magnetic COF sorbent using two novel monomers of *1,2,4,5*-Tetrakis-(*4*-formylphenyl) benzene (TFPB) and *p*-Phenylenediamine (PPD) at room temperature (Figure 1b). The adsorption capacities for AFs ranged from 69.5 to 92.2 mg g^−1^. Under the optimized conditions, the SPE extraction efficiency was enhanced, saving both time (5 min) and organic reagent (2 mL), and satisfactory results for AF detection in food matrices (milk, edible oil, and rice) were obtained. The magnetic COF sorbent can be reused more than eight times. Wei et al. [75] developed a vortex-assisted magnetic SPE method, using a core–shell structured magnetic covalent organic skeleton (FeO/COF-TpBD) as the adsorbent for rapid and simultaneous extraction of ten mycotoxins commonly found in maize (Figure 1c). The prepared magnetic adsorbent was demonstrated to have the advantages of strong magnetism and good stability, which also obtained high sensitivity (LOD: 0.02–1.67 μg kg^−1^) and recovery (73.8–105.3%). Furthermore, the adsorbent dosage (5 mg) and required time (0.5 min each for adsorption and desorption) were greatly shortened compared with previous reports. Due to the complexity of the food matrices and the rapid consumption of food, magnetic SPE sorbents are required to have good chemical stability, strong dispersion ability, and a high affinity for mycotoxins. These attributes are crucial for ensuring the efficiency and reproducibility of magnetic SPE purification process. Therefore, magnetic SPE adsorbents used for food matrices’ purification are typically designed as core–shell structures. That means functionalizing the surface of the magnetic core to form a specific recognition shell with high affinity for the targets. The preparation of such core–shell magnetic SPE adsorbents involved multiple complex steps, leading to significant batch-to-batch variations, which limited the widespread application of these materials to some extent. Additionally, the magnetic properties may decrease after multiple modifications on the surface of magnetic nanoparticle core, directly affecting the efficiency of adsorption and separation. Therefore, it is necessary to develop magnetic SPE materials that are easy to prepare, possess stable magnetic properties, and exhibit a high affinity for the target compounds.

### 2.2. Absorbent for SPME 

In contrast to traditional SPE technology, SPME greatly simplifies the analytical operation procedure, reduces the extraction time, and enhances the extraction efficiency, and it has been emphasized in food detection [76,77,78]. Nanomaterial-based novel solid-phase adsorbent materials possess a larger specific surface area, suitable pore size, and surface structure, along with excellent adsorption and mechanical properties [79,80]. These features enable the highly selective adsorption of target analytes in complex matrices, showing great potential for application in SPME [81]. This provides crucial support for the rapid and highly sensitive detection of mycotoxins.

Gold nanoparticles (AuNPs) are composed of nanoscale gold atoms (1–100 nm), which not only have unique optical, electrical, and excellent surface enhancement properties but also have a high specific surface area. AuNPs can be used as the carrier of Abs, aptamers, and other specific recognition molecules, making them the most commonly used material in the field of food detection [82,83,84]. Zhang et al. proposed an innovative approach for the facile and controllable preparation of an aptamer-functionalized capillary monolithic polymer hybrid, which achieved high specificity and high affinity for the determination of patulin in food samples [43]. AuNPs with patulin aptamers were directly modified via Au-S bonds on capillary monolithic columns (Figure 2a). These aptamer-functionalized capillary monomer–polymer hybrid materials were applied as the SPME adsorbent combined UPLC-MS/MS to achieve very high sensitivity and selectivity for patulin, with an LOD of 2.17 pmol L^−1^ and linear range of 0.0081–8.11 nmol L^−1^. Based on the co-polymerization reaction of methacrylic acid and divinylbenzene, Wu et al. developed a poly (methacrylic acid-co-divinyl-benzene) [poly (MAA-co-DVB)] monolithic column for the in-tube SPME of three mycotoxins [44]. The high-strength micro/nanostructure formed by two polymer monomers contained a large number of acrylic acid groups capable of forming hydrogen bonds with the carbonyl, hydroxyl, and hydrophobic benzene groups in the structure of AFB_1_, ZEN, and sterigmatocystin. Therefore, the developed monolithic column based on poly (MAA-co-DVB) had a high recognition ability for three target molecules, effectively overcame the matrix effect of rice food, and realized the highly sensitive determination of three target mycotoxins.

Dispersive solid-phase microextraction (DSPME) is a non-fiber extraction method, which operates by mixing the solid-phase sorbent directly with the sample medium through vortexing and sonication [85,86,87]. After target adsorption, the sorbent was separated and desorbed the target with a suitable solvent for further detection and analysis. This approach eliminates the need to coat the adsorbent onto a carrier surface, thereby increasing the contact area between the adsorbent and the sample solution, leading to improved extraction efficiency and reduced adsorption time. However, due to the mixing of the sorbent with the sample matrix, the separation process becomes challenging [88,89]. Therefore, magnetically separable magnetic adsorbent materials have attracted significant attention from researchers. The reverse-phase/phenylboronic-acid (RP/PBA) magnetic microsphere prepared by Xu et al. demonstrated rapid and efficient dispersive extraction of amatoxins and phallotoxins [90]. The phenyl and phenylboronic-acid groups on the surface allowed for selective adsorption of the target toxins through hydrophobic interaction, π-π, and boronic acid affinity, thereby significantly reducing matrix effects in UPLC-MS/MS analysis. The proposed method obtained satisfactory linearity (r > 0.9930), LOD (0.3 μg kg^−1^), and recovery (97.6–114.2%) values. Javier and co-workers synthesized core–shell poly (dopamine) magnetic nanoparticles, which were used for the simultaneous extraction of six mycotoxins (zearalenone, α-zearalanol, β-zearalanol, α-zearalenol, β-zearalenol, and zearalanone) from milk products [91]. This DSPME method combined with magnetic separation provided a very promising scheme for the rapid extraction and high-throughput detection of mycotoxins in food substrates. 

Carbon-based nanomaterials, MOFs, and MIPs are some of the commonly used materials in SPME [92,93,94]. They have unique nanostructures and excellent physical and chemical properties, which help to improve the analytical efficiency, selectivity, and sensitivity of complex matrix extraction processes, and are widely used in the detection of mycotoxins [95,96]. Graphene is a typical carbon-based nanomaterial with an abundance of oxygen-containing functional groups (-OH and -COOH) on the surface. These functional groups can form hydrogen bonds or electrostatic interactions with target molecules, thus facilitating the adsorption process [97,98]. In addition, the modification of the graphene surface can reduce the agglomeration phenomenon and enhance its adsorption capacity. The group of Wu et al. prepared reduced rGO and ZnO nanocomposites (rGO-ZnO) through a hydrothermal process for separation, purification, and enrichment of 12 mycotoxins [99]. The key parameters affecting DSPME, including the extraction solution, eluent, and dosage of adsorbent, were optimized in detail to obtain the ideal purification and extraction efficiency. In combination with UHPLC-MS/MS, the prepared rGO-ZnO material was applied for extraction and analysis of 12 mycotoxin targets, achieving high sensitivity (LOQ: 0.09–0.41 µg kg^−1^) and satisfactory precision (RSD: 1.4–15.0%). MOF materials have extremely high porosity, excellent thermal stability, and a large specific surface area. Their tunable pore size, porous channels, and nano-space make them ideal SPME adsorbent materials. Lotfipour and co-workers [46,47] prepared a vitamin-based MOF material, using vitamin B_3_ as a bio-linker and cobalt ions as a metallic center in water, and applied it as a sorbent in a DSPME of patulin and Ochratoxin A (OTA) from fruit juice samples and four aflatoxins (AFB_1_, AFB_2_, AFG_1_, and AFG_2_) from soy milk (Figure 2b). High extraction efficiencies can be achieved by mixing the precipitated protein supernatant with the sorbent and simply vortexing and centrifuging. This MOF material exhibited an excellent adsorption capacity for the target mycotoxins and can be prepared on a large scale in an environmentally friendly manner. The developed strategy required only a small amount of sorbent and organic solvent during the extraction process, which was one of its significant advantages. Using the microfluidic self-assembly technology, Wang et al. successfully prepared magnetic inverse photonic microspheres with a regular three-dimensional ordered macroporous structure and further utilized the “virtual template” molecular imprinting method to fabricate an MIP with high selectivity [100]. This MIP material, with the advantages of an adjustable pore size, easy modification, and good thermal stability of photonic crystal microspheres, was used as a DSPME adsorbent in combination with HPLC to realize the rapid quantitative analysis of AFB_1_.

## 3. Nanoscale Materials for Rapid Detection and Screening

In view of the rapid consumption of food, the detection of food contaminates (including mycotoxins) needs to be rapid, high-throughput, and able to meet the requirements of in situ detection and easy popularization. Although highly sophisticated large-scale analytical instruments can provide accurate, sensitive, and reproducible strategies for the detection of mycotoxins in complex food matrices, they often involve complex and cumbersome sample preparation processes and require experienced operators, which present significant drawbacks in terms of cost, convenience, and popularity. In contrast, immune, aptamer, or sensing assays based on the specific recognition or binding of Abs, aptamers, or artificial Abs made up for the above shortcomings and became powerful tools for the rapid screening and detection of mycotoxin contamination [101,102]. Various nanoscale materials with special functions and properties are constantly being developed as carriers of biomolecules, signal sources, or other applications for the development of novel strategies or devices for the rapid detection of mycotoxins [103,104,105]. Table 3 illustrates the application of various nanoscale materials for the rapid detection and screening of mycotoxins in food substrates.

### 3.1. Carrier for Biometric Molecules

Carbon-, metal-, and semiconductor-based nanoscale materials often have large specific surface areas and excellent optical or chemical properties and are inherently rich in binding sites or can be modified to obtain them [144,145]. These binding sites can stably immobilize recognition molecules such as Abs or aptamers on the surface of materials and maintain their biological activity, thus improving the sensitivity and stability of detecting targets. The Abs or aptamers loaded on the nanomaterials specifically bind to the targets, which also provide a basis for the highly sensitive and stable detection of mycotoxins. The key to the research lies in selecting suitable nanoscale materials and functionalization methods to ensure the stable immobilization of Abs or aptamers with specific recognition ability [146].

An immunochromatographic assay (ICA) is a rapid, portable, and user-friendly immunodiagnostic tool that has been extensively studied [147,148]. ICAs generally use nanomaterials (e.g., AuNPs, magnetic NPs, quantum dots (QDs), etc.) to label specific Abs or antigens and bind to targets in the test sample and undergo chromatographic separation through capillary action on the test strip, resulting in the appearance of different color bands at specific locations [149,150,151]. Rapid screening and semi-quantitative analysis of target molecules can be realized by visual reading or simple color-reading instruments. ICAs not only have high sensitivity and specificity but also do not require complicated instruments and can produce results in a few minutes, features that are important for rapid screening and real-time detection. In the study of Dzantiev et al. [152], a scheme for competitive ICA with indirect labeling was implemented and developed for the detection of ZEN in baby food. Two separate reagents, free specific Abs and anti-species Abs conjugated with AuNPs (30 nm), were used to improve the sensitivity and the reliability of measurements at the same time. The LOD for ZEN in baby food was 5 pg mL^−1^ (100 pg g^−1^), and the time required for the entire analysis was only 17 min. Li and Zhang’s study introduced a novel time-resolved fluorescence ICA (TRFICA) using two idiotypic nanobodies for the simultaneous detection of AFB_1_ and ZEN [153]. Enhanced fluorescent Eu/Tb (III) nanospheres were employed as labels, which were conjugated to the anti-idiotypic nanobody (AIdnb) and monoclonal antibody (mAb). Based on this, they developed and compared two competitive TRFICAs, namely AIdnb-TRFICA and mAb-TRFICA (Figure 3a). The AIdnb-TRFICA exhibited lower semi-inhibitory concentrations for AFB_1_ and ZEN (0.46 and 0.86 ng mL^−1^, respectively) and was 18.3-fold and 20.3-fold more sensitive than mAb-TRFICA. This is the first report about a time-resolved strip method based on AIdnbs for dual mycotoxins. Du et al. [154] proposed an enhanced ICA strip for a simultaneous semi-quantitative and quantitative detection of OTA and AFB_1_ by using a signal strategy based on gold growth on the surface of E. coli K12 carrier (Figure 3b). The larger surface area, better biocompatibility, and high loading capacity of the E. coli K12 carrier played an important role in improving the performance of ICAs that have a visual semi-quantitative LOD of 0.03 ng mL^−1^, which was 17-fold and 33-fold lower than conventional ICA strips. This study provided an important reference for sensitive, simultaneous, rapid, and in situ monitoring of multicomponent contaminants. QD microspheres (QDMs) are usually a few nanometers to tens of nanometers in size [155,156]. After surface modification, the Abs can be stably immobilized on the surface by chemical cross-linking, covalent binding, or affinity binding, which maintains the recognition ability of Abs, thereby improving the detection accuracy and sensitivity. On the other hand, QDM can immobilize multiple Abs simultaneously, enabling multiple identification and a multi-parameter analysis, bringing more potential and flexibility for further application of methods such as ICAs [157,158]. Wang and Liu [159] immobilized specific Abs against AFB_1_, OTA, and ZEN toxins on the surface of QDM and designed an immunochromatographic test strip capable of simultaneously detecting the three toxins (Figure 3c). When applied to grain samples, the established method exhibited high accuracy, sensitivity, and remarkably low LODs (AFB_1_, 0.01 ng mL^−1^; OTA, 0.2 ng mL^−1^; and ZEN, 0.032 ng mL^−1^). Notably, the proposed multiple toxin detection strategy required only simple manual operation and one portable handheld strip reader, enabling on-site testing to be completed within 45 min. This represents a valuable solution for rapid, sensitive, and on-site detection of multiple mycotoxins. 

Aptamers are short deoxyribonucleic acid (DNA) or ribonucleic acid (RNA) oligonucleotide sequences with high affinity which show excellent specificity for various target molecules, such as proteins, DNA, small molecules, and metal ions [160,161]. As recognition elements of mycotoxins detection, they have become strong competitors against Abs [162,163,164]. Sun and Xie developed a novel enhanced enzyme-linked aptamer assay (ELAA) for the detection of ZEN based on AuNPs modified with aptamers and horse radish peroxidase (HRP) [165]. AuNPs with a high specific surface area were utilized as a carrier to immobilize more aptamer probes labeled with HRP, thus enhancing the catalytic ability of HRP to amplify the colorimetric signal and improve the detection sensitivity of ZEN. In the study of Wang et al. [166], a turn-on aptasensor for the simultaneous time-resolved fluorometric determination of ZEN, trichothecenes (T-2), and AFB_1_ was designed and developed. Multicolor-emissive NPs doped with lanthanide ions (Dy^3+^, Tb^3+^, and Eu^3+^) were functionalized with specific aptamers as bio-probes, and the tungsten disulfide (WS2) nanosheets were applied as the fluorescence quencher. Three toxin targets can be identified in a single solution simultaneously without mutual interference. Under the same excitation wavelength (273 nm), the LODs for three toxins were 0.51 (ZEN), 0.33 (T-2), and 0.40 pg mL^−1^ (AFB_1_), respectively, demonstrating the remarkable sensitivity. This strategy provided an advanced guidance for the rapid quantitative analysis of multi-toxin targets in a single sample. Through computational simulations, Bagherzadeh and co-workers [167] designed two novel highly functional mycotoxin aptamers (F20 and F20-T), which were used as recognition elements in the lateral flow aptasensors (Figure 3d). The two selected aptamers showed very good affinity for multiple mycotoxins (AFB_1_, AFM_1_, AFG_1_, AFG_2_, OTA, and ZEN). After coupling with AuNPs, the developed two aptamer-AuNPs test strips working in competitive mode showed high sensitivity. The F20-T-based strips (LOD: 0.1 ng mL^−1^) were more sensitive than those utilizing F20 (LOD: 0.5 ng mL^−1^). Utilizing a simple strip reader, both of the developed strips were able to analyze AFB_1_ in maize flour within 30 min.
Figure 3(**a**) Time-resolved fluorescence ICA using idiotypic nanobodies to detect AFB_1_ and ZEN [153]. Copyright Analytical Chemistry, 2017. (**b**) Schematic diagram of enhanced ICA strip based on gold growth on the surface of *E. coli* carrier for mycotoxins detection [154]. Copyright Talanta, 2023. (**c**) Sample pretreatment and QDM probe preparation (**A**) and target detections (**B**). [159]. Copyright Food Chemistry, 2022. (**d**) Aptamer-based lateral flow strip for AFB_1_ detection in the presence or absence of AFB_1_ [167]. Copyright Food Control, 2023.
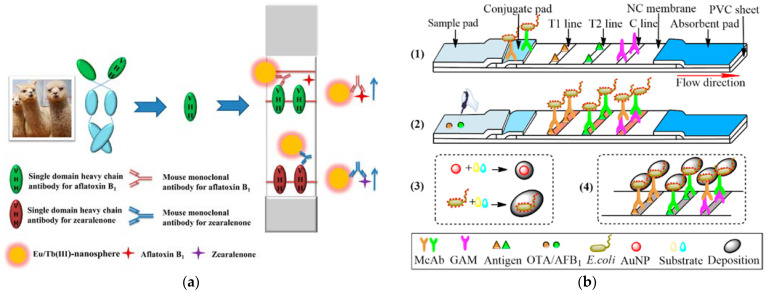

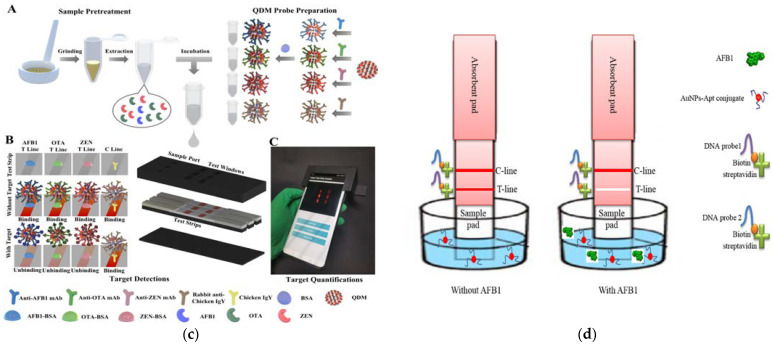


### 3.2. Signal Source for Rapid Detection

Size effect of nanoscale materials endows them with unique optical, electrical, magnetic, and mechanical properties that enable them to generate signals with specific characteristics, providing a signal source for the development of highly sensitive and high-performance mycotoxin detection strategies [168,169]. The given large specific surface area allows the material to fully engage and interact with the surroundings, thus enhancing the signal generation capability of nanomaterials and enhancing the interaction efficiency with other substances.

The size of QDs is close to the wavelength of electrons, which results in typical quantum confinement and quantum size effects. These effects lead to QDs having optoelectronic properties that are different from those of conventional materials, such as typical luminescence dispersion properties, a long luminescence lifetime, a wide luminescence spectral range, and high fluorescence quantum efficiency, making them highly preferred in optical detection [170,171,172]. The research of Solanki reported the first graphene QDs-based fluorescence sensor for AFB_1_ [173]. Through non-radiative resonance energy transfer between the graphene QDs and AFB_1_ molecules, AFB_1_ directly acted as the quencher of graphene QD fluorescence. Notably, this innovative sensor achieved direct quantification of AFB_1_ without the use of inhibitors or biometric elements and obtained an acceptable linear response range of 5-800 ng mL^−1^ with an LOD of 0.158 ng mL^−1^. Nanosized MOFs (NH_2_-MIL-53 (Al)) can undergo hydrolysis in alkaline environments and release a large number of fluorescent ligands (NH_2_ center dot H_2_BDC). This property allowed it to be applied as large-capacity nanovesicles for loading signal molecules to investigate various biorecognition processes. Based on this, Fu et al. [174]. developed a valuable immunoassay for AFB_1_ by using NH_2_-MIL-53 (Al) as a fluorescent signal probe (Figure 4a). Using a competitive immunoassay mode, on a microplate, AFB_1_ can be analyzed in a linear range of 0.05–25 ng mL^−1^. This method demonstrated significant advantages, such as high sensitivity, simplicity of operation, and ideal reliability, making it suitable for application in aptasensors and receptor binding assays. Zhang et al. [123] presented a pioneering study on a unique fluorescent sensing platform for detecting AFB_1_ by utilizing an extremely low concentration of Al-MOFs (Figure 4b). Remarkably, due to the fluorescence source of Al-MOF, this sensor exhibited the largest fluorescence enhancement (or quenching) constant of 179,404 M^−1^ among all reported luminescent chemosensors. Based on this, the sensor enabled a rapid and precise quantitative analysis of the target AFB_1_, with a linear response in the concentration range of 0.05–9.61 μM and an LOD as low as 11.67 ppb. This research represented a significant advancement in the field of MOF-based fluorescence detection and offered promising prospects for future applications.

Förster Resonance Energy Transfer (FRET) is a non-radiative energy transfer process that is based on a confined radiation-free electric dipole coupling. Energy transfer between nanomaterials can also be achieved through FRET, usually involving two different nanostructures or NPs [175,176]. When the excited state of the energy donor NPs overlaps with the ground state of the energy acceptor NPs and the distance between them is close enough (usually 1-10 nm), the energy of the donor can be transferred to the acceptor. Using this FRET transfer mechanism, fluorescent probes based on nanoscale materials were constructed for fluorescent sensing to detect mycotoxins [177,178,179]. Wang et al. proposed a turn-on luminescence RET strategy for OTA detection in beer [180], which applied the upconversion NPs (UCNPs) modified by biotin-labeled OTA aptamers of the type NaYF_4_: Yb, with Er as energy donor and Au nanorods (Au NRs) modified by thiolated OTA aptamer cDNA as the acceptor. When excited under a 980 nm laser, the emission peak of UCNPs at 657 nm overlapped with the absorption peak of AuNRs at 660 nm. Through the hybridization of nucleic acid sequences modified on the surface of UCNPs and AuNRs, the distance between them is shortened, leading to the quenching of luminescence. Mo et al. reported on a WS2 nanosheet sensing platform based on chemiluminescence RET for OTA detection [181]. The OTA aptamer-12-mer linker-G-Quadruplexes/Hemin DNAzymes-H_2_O_2_-luminol and WS2 nanosheets acted as the energy donors and acceptors, respectively. This chemiluminescence RET sensing system worked based on changes of chemiluminescence intensity caused by the presence or absence of OTA-induced donor affinity or acceptor surface desorption. It is very meaningful to modify the recognition sequence only by using the substrate adapter that can be easily extended to the analysis of other targets.

Due to the tunable particle size, optical properties, high stability, and controllability, AuNPs were also extensively employed in signal transduction and enhancement studies [182,183]. AuNPs of different sizes, shapes, and aggregation states exhibited colors from red to purple or blue, which have been widely applied in colorimetric analysis [184]. Recently, a series of target-induced AuNPs’ aggregation-based colorimetric analysis methods were developed. These methods relied on the dispersion and aggregation of AuNPs under high salt conditions. Specifically, salt ions induce the aggregation of AuNPs, and ligands protect AuNPs from aggregation. When the target molecule was introduced, the specific interaction with the ligands led to the change of AuNPs’ structure, resulting in the loss of their protective effect. As the degree of NPs’ aggregation increased, the absorption wavelength changed, resulting in a blue-colored solution. Based on this principle, rapid detection methods for AFB_2_, OTA, and AFB_1_ were developed [185,186]. Mehdi Dadmehr and co-workers utilized carboxyfluorescein (FAM) tags to label specific aptamers with internal complementary sequences [187]. This aptamer was further modified onto the surface of a GO/AuNPs nanocomposite to form a heterogenous double-stranded stem–loop structure, resulting in fluorescence quenching. In the presence of AFB_1_, the aptamer formed a complex with AFB_1_, leading to the denaturation of the stem–loop structure back to the single-strand, thereby recovering the fluorescence signal. This strategy was significantly more selective for AFB_1_ than other analytes and responded well to AFB_1_ concentrations in the range of 0.5–20 pg mL^−1^ with a low LOD (0.1 pg mL^−1^). A similar principle has been applied in T-2 toxin detection, using a Cu MOF material as a fluorescent quencher with dual-sided FAM labeling [188].

### 3.3. Other Functions for Mycotoxin Detection

The unique size and structure of nanoscale materials allowed them to exhibit special electrical and optical properties. These distinct characteristics, different from those of macroscopic materials, have expanded the application of nanoscale materials in various fields and provided new research directions for food safety detection, especially mycotoxin detection [189,190].

Electrochemical sensors have the advantages of a simple operation, high sensitivity, low cost, and easy miniaturization, making them highly suitable for the rapid screening and on-site detection of food samples on a large scale [191]. Modification of the surface of electrochemical transducers with various functional nanoscale materials can be used for the immobilization of recognition molecules, signal amplification, or the enhancement of detection stability, thereby becoming an effective method to solve the problems of detection specificity and sensitivity in electrochemical strategies for food samples [192,193]. MWCNTs offered the properties of high electrical conductivity, biocompatibility, a high surface-to-volume ratio, and chemical stability, which employed an ideal and promising material for electrochemical application [194,195]. Bai and his colleagues [196] established a novel aptasensor based on MWCNTs, graphene, and a target-induced amplification strategy for the detection of ZEN (Figure 5b). The prepared chitosan-functionalized acetylene black@MWCNTs nanocomposites have a large specific surface area, good conductivity, and a remarkable film-forming ability, which not only increased the immobilization of aptamers but also improved the stability of aptamer sensors and can be used to label aptamers that specifically recognize ZEN and enhance the intensity of signal changes. This effective strategy can be applied to the analysis of other targets by ligand substitution. Zhai et al. [197] functionalized Ce-based MOF and MWCNTs nanocomposites with polyethyleneimine and used them to construct an electrochemical sensing platform with a high surface area and high electrochemical activity (Figure 5c). This study employed Pt@Au NPs for ZEN-specific aptamer attachment, improving the detection sensitivity and responsiveness of the sensing platform. Arben Merkoci’s team designed an inkjet-printable water-based GO ink and prepared working microelectrodes via electrochemical reduction [198]. The microelectrodes were conjugated with HT-2 toxin antigen-binding antibody fragments for immunosensing HT-2 toxin, obtaining a lower LOD (1.6 ng mL^−1^). This GO-based printed electrode has the advantages of convenient large-scale preparation, a low cost, and broad application prospects. 

The electrochemical strategy has also been used for the preparation of novel NPs and the construction of new detection devices during the process of toxin detection. Based on 3D-graphene and iron nanoflorets, Saheed et al. constructed a “Dandelion” nanostructure (iron nanoflorets on 3D-graphene–nickel) as the transducer to develop a sensitive biosensing system [199]. The study first prepared graphene with a 3D structure via low-pressure chemical vapor deposition and then further electrochemically deposited iron nanoflorets. The “Dandelion” nanostructure has a large surface area for effective and feasible bio-capture and has a high stability, with 30.65% activity still maintained after 48 h. The constructed biosensor showed remarkable discrimination of deoxynivalenol (DON), with an LOD of 2.11 pg mL^−1^, providing an efficient, simple, and economical solution for DON detection in food and feed samples. Polyaniline (PANI) is a semi-flexible conducting polymer that is environmentally friendly, stable, inexpensive, and biocompatible and has excellent conductivity [200,201]. During the MIP formation, PANI can bind to the template molecule through intermolecular interactions such as hydrogen bonding and π-π interactions; more important, for MIPs, the template molecules can easily be removed from the PANI polymer. Solanki et al. [202] directly electrophoretically deposited PANI-based MIPs films on ITO-coated glass substrates (Figure 5d) and further used differential pulse voltammetry (DPV) to detect AFB_1_ and FuB_1_ mycotoxins, with LODs of 0.313 and 0.322 pg mL^−1^, respectively.
Figure 5(**a**) Design of the nano-aptasensing platform anti-AFM_1_/FcTGL/AuNPs/SPCE [110]. Copyright Microchemical Journal, 2021. (**b**) Preparation of the CGO-ZBA bioconjugate (**A**) and the fabrication of ZEN aptasensor (**B**) [196]. Copyright Food Chemistry, 2021. (**c**) Synthesis of P-Ce-MOF@MWCNTs and stepwise construction of the electrochemical aptasensor [197]. Copyright Food Chemistry, 2023. (**d**) Molecularly imprinted electrochemical platform for AFB_1_ and FuB_1_ detection [202]. Copyright Microchemical Journal, 2021.
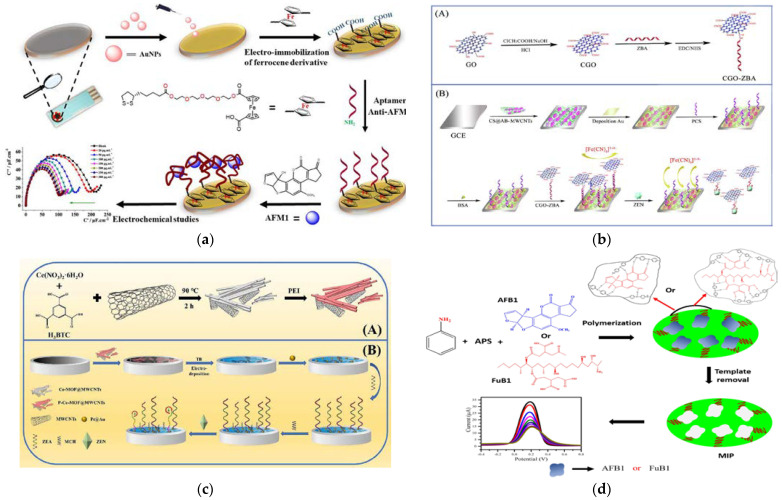


The composition, shape, and size of nanomaterials can regulate their ability to absorb and scatter light, especially noble metals such as Au, Ag, and Pt. When light strikes a noble metal NP, if the frequency of the photon matches the collective oscillation frequency of the electrons in the NP, the NP will strongly absorb the energy of the photon, resulting in a phenomenon known as localized surface plasmon resonance (LSPR) [203,204]. Although the LSPR-based label-free sensing technique has the advantages of simplicity, sensitivity, and a low cost, the in situ detection of toxin targets in real samples is still challenging, which is mainly due to the instability of NPs in specific pH and salt concentration environments. Chip-based LSPR sensors can address this issue by directly immobilizing on substrates, but it is very different to precisely control the sample volume and accurately load the sample onto the chip. Fiber-optic-based LSPR sensors can solve the above problems, laying the foundation for designing portable, simple, and highly sensitive sensors. Kim et al. [136] reported an LSPR-sensitized detection system based on optical fibers and Au nanorods (Figure 6a). An OTA-specific aptamer on Au nanorods was directly coated on the surface of the optical fiber as the sensing probes. The detection process needed only to immerse the optical fiber in the sample extraction solution. The rapid and sensitive detection of OTA was achieved by monitoring the change of the LSPR peak induced by aptamers on the Au nanorods specifically recognizing and capturing the target. The above method intelligently combined extraction, enrichment, and detection into a simplified process, achieving a nondestructive, simple, and sensitive detection of the target toxin. Yan et al. successfully constructed an LSPR-based optical fiber tip facet biosensor involving the sequential assembly of -NH_2_ functional groups, AuNPs, and FB_1_-specific aptamers on the fiber end face [205]. AuNPs not only provided an enhanced LSPR effect but also served as stable carriers for specific sequences. The LSPR peak shift of the proposed fiber end-face aptasensor showed a good linear relationship with the logarithmic concentration of FB_1_ (R^2^ = 0.9817), and the LOD was 0.17 ng mL^−1^. Jaroon’s research team designed an unlabeled colorimetric aptasensor, which exploited the selective interaction between the aptamer and AFM_1_, leading to structural changes while allowing for the aggregation of AuNPs induced by NaCl [206]. The solution’s color changes resulting from the LSPR peak shift of the aggregated AuNPs can be directly quantitatively measured using colorimetry. The sensor exhibited outstanding analytical sensitivity (LOD: 0.002 ng mL^−1^), meeting the maximum residue limit requirement for AFM_1_.

Photonic crystal microspheres (PCMs) are microsized particles with an internal photonic crystal structure, possessing diverse optical properties, including abundant optical modes and optical resonances [207]. In a PCM suspension assay, the surface of PCMs was modified with various biorecognition molecules (Abs, aptamer, etc.) to capture specific targets. When the target was bound with the appropriate biorecognition molecules, the optical properties of PCMs changed, enabling rapid and accurate detection of the presence and quantity of the target through an optical detection system [208,209]. Zheng and co-workers [210] developed a PCM suspension array based on aptamer fluorescence recovery for the detection of multiple mycotoxins in grain samples (Figure 6b). Fluorescent dye and quencher-labeled toxin aptamer and anti-aptamer hybrid double-stranded DNA were first immobilized on the PCM surface. When different amounts of toxin targets were combined with aptamers, the fluorescence of PCMs recovered to varying degrees, and different toxins could be distinguished by observing the color change. The fluorescence signal intensity of PCMs obtained in this study was nearly 100-fold higher than that of solid glass beads, offering the advantages of ultra-sensitivity to AFB_1_, OTA, and FB_1_; high selectivity; and small reagent usage. Li et al. [211] established a multiple surface enhancement of Raman scattering (SERS) array for mycotoxins based on AuNPs-loaded inverse opal silica PCM (SIPCM) (Figure 6c). The array had very high detection sensitivity and could be reused after regeneration. Different Abs and organic dyes were covalently or noncovalently grafted onto the surface of AuNPs as Raman tags to encode different toxins. Hot spots were generated when the Raman tags were close to the AuNPs immobilized on the surface of the SIPCM. The SERS enhancement effect of this AuNPs dimer-based SIPCM could be up to 8-fold. Three Raman nanotags can be integrated into an array, relying on the competition with the antigen immobilized on the surface of SIPCM, which can be used for the qualitative and quantitative analysis of OTA, FB_1_, and DON (three toxins) simultaneously. This AuNPs-based SERS assay possessed a wide linear concentration range and a low LOD, which enabled it to rapidly and simultaneously screen multiple mycotoxins in real samples, giving it a great potential for application.
Figure 6(**a**) Detection process of LSPR aptamer sensor based on optical fiber (**A**) and aptamer-modified GNRs are chemically attached to the optical fiber core (**B**) and An LSPR peak shift induced by OTA binding (**C**). [136]. Copyright Biosensors and Bioelectronics, 2018. (**b**) Aptamer fluorescence signal recovery screening for multiplex mycotoxins in cereal samples based on photonic crystal microsphere suspension array [210]. Copyright Sensors and Actuators B: Chemical, 2017. (**c**) The principle of multiplex mycotoxins SERS immunoassay on AuNPs-loaded SIPCMs [211]. Copyright Sensors and Actuators B: Chemical, 2021.
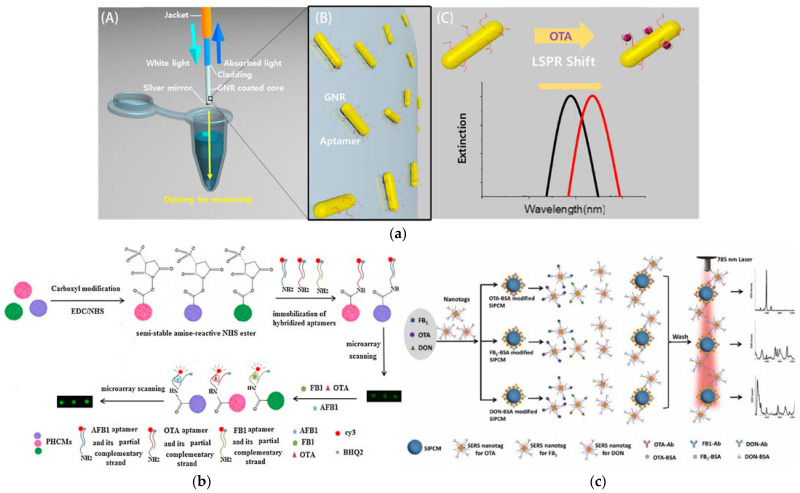


## 4. Conclusions and Perspectives

In recent years, significant advancements have been made in the field of mycotoxin detection in food due to the utilization of various nanoscale materials. These materials have enabled the development of rapid, precise, sensitive, and cost-effective detection methods. Among the diverse array of nanoscale materials, rigid nanomaterials stand out due to their exceptional characteristics, including a high specific surface area, substantial loading capacity, and stability. These attributes make them particularly advantageous in the development of materials for solid-phase extraction (SPE) and solid-phase microextraction (SPME) pretreatment, enhancing the performance of purification processes for precise instrument analysis. Furthermore, flexible nanomaterials, characterized by excellent biocompatibility and specific optoelectronic properties, introduce a new dimension to the rapid mycotoxin detection landscape. They facilitate the preservation of the activity of biorecognition molecules and signal sources, improving the overall efficiency of mycotoxin detection strategies. Despite these valuable advances, several challenges remain. Firstly, nanoscale materials with better properties are still needed, both for efficient and stable pretreatment process and high-throughput rapid detection process. Among them, green synthesis and the high-volume production process of nanoscale materials require more in-depth research. Secondly, there is an urgent need to develop automated and miniaturized detection equipment or devices for mycotoxins in food based on nanomaterials. Such devices would streamline the detection process, minimize operational errors, and promote the widespread adoption of cost-effective detection strategies. Overall, nanoscale materials provide opportunities, as well as challenges, for research on the detection and monitoring of mycotoxins in food.

## Figures and Tables

**Figure 1 foods-12-03448-f001:**
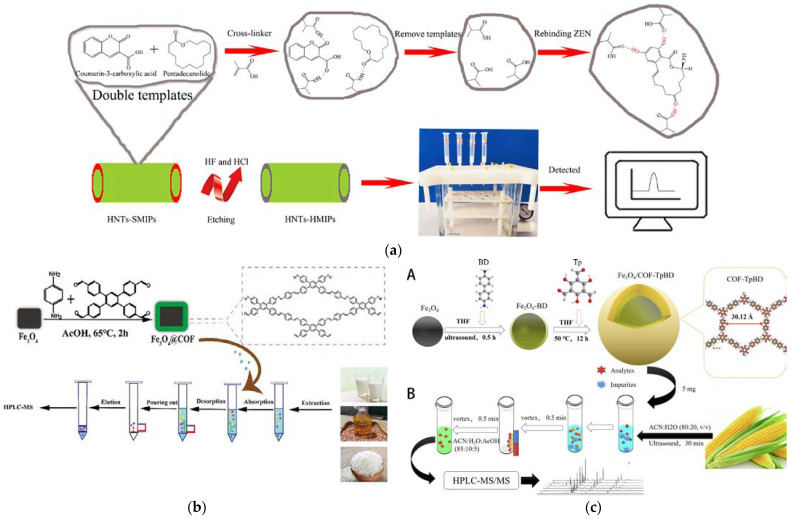
(**a**) Preparation route of HNTs-HMIPs [42]. Copyright Microchemical Journal, 2020. (**b**) Magnetic COF sorbents as sorbents for SPE of AFs in food [74]. Copyright Food Chemistry, 2022. (**c**) Fe3O4/COF-TpBD preparation (A) and the magnetic solid phase extraction application for the determination of the ten mycotoxins (B). [75]. Copyright Food Chemistry, 2023.

**Figure 2 foods-12-03448-f002:**
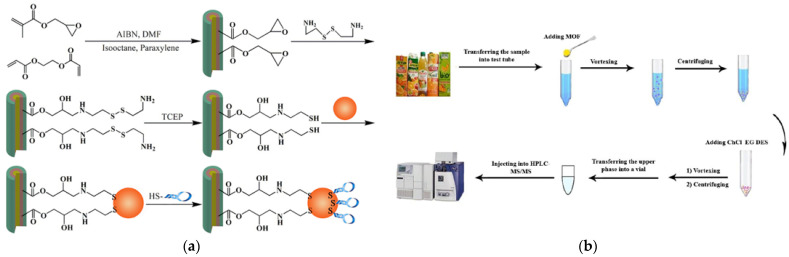
(**a**) Preparation of the aptamer-functionalized capillary monolithic column for the recognition of patulin [43]. Copyright Food Control, 2021. (**b**) Green-synthesized MOF-based DSPME procedure for the determination of mycotoxin in juice [46]. Copyright Microchemical Journal, 2022.

**Figure 4 foods-12-03448-f004:**
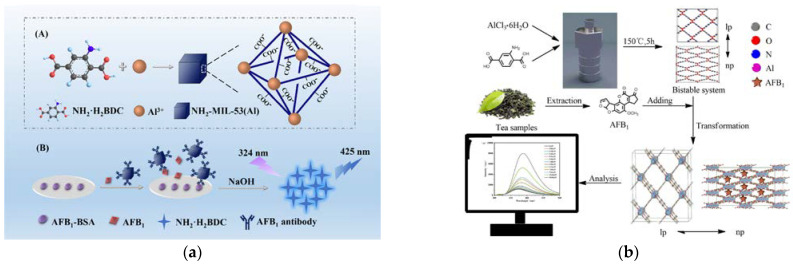
(**a**) Preparation of MOFs NH_2_-MIL-53(Al) (A) and illustration of competitive FIA of AFB_1_ (B) [174]. Copyright ACS Applied Materials & Interfaces, 2019. (**b**) Fluorescent detection of aflatoxin B_1_ based on fluorescent MOFs [123]. Copyright Food Chemistry, 2022.

**Table 1 foods-12-03448-t001:** Maximum permissible limits for mycotoxins in foods of different countries or organizations.

**The United States**	Total amount of AFB in food: <20 μg/kg; DON: <1000pg/kg, ZEN: <100 pg/kg; Milk and dairy products: AFM_1_ ≤ 0.5 μg/kg.
**European Union**	Agricultural products: Total amount of AFs: <4 μg/kg, AFB_1_: <2 μg/kg, OTA: <3 μg/kg, DON: <1000 μg/kg, ZEN: <50 μg/kg; Infant foods: Total amount of AFB: <2 μg/kg, AFB_1_ <0.1 μg/kg, AFM_1_: <0.025 μg/kg, OTA: <0.5 μg/kg, DON: <150 μg/kg, ZEN: <20 μg/kg
**China**	Corn, peanuts, and their products: AFB_1_: < 20 μg/kg, OTA: <5 μg/kg, DON: <1000 μg/kg, ZEN < 60 μg/kg; Other grains, beans, and fermented foods: AFB_1_: <5 μg/kg; Infant foods: AFB_1_: 5 μg/kg, AFM_1_: < 0.5μg/kg; Fresh milk and dairy products: AFM_1_: < 0.5μg/kg; Rice and vegetable oils (except corn oil and peanut oil): AFB_1_: <10 μg/kg.
**Japan**	Peanuts and their products: AFB_1_: <10 μg/kg; Wheat: DON: <1100 μg/kg; Apple juice: Patulin: <50 μg/kg.

**Table 2 foods-12-03448-t002:** Application of various nanoscale materials in SPE and SPME processes for the detection of mycotoxin in food.

Materials/Methods	Mycotoxins	Substrates	Properties of Materials	Results	Ref.
**SPE**					
PDA-IL-NFsM SPE coupled with UPLC-MS/MS	AFB_1_, AFB_2_, AFG_1_, AFG_2_, ST, FB_1_, FB_2_, OTA, ZEN, HT-2, T-2, DON, 3-AcDON, NIV, 15-AcDON	Corn, wheat	Various interception mechanisms with the target through hydrogen bonding, π-π interaction, and electrostatic or hydrophobic interaction; good simultaneous adsorption performance; significantly reducing the matrix effect	Linear range: 1.0–2000 μg/kg; LOD: 0.04–4.21 μg/kg; LOQ: 0.13–14.03 μg/kg; Recovery: 80.79–112.37 % (RSD: 2.91–14.82 %, *n* = 4)	[33]
Fe_3_O_4_@COF Magnetic SPE coupled with UHPLC-MS/MS	AFB_1_, OTA, ZEN, TEN, ALT, ALS, AME, AOH, TEA	Fruits	Abundant aromatic rings and carbonyl groups in Fe_3_O_4_@COF structure; through the strong π-π interaction and hydrogen bond between mycotoxin and Fe to realize effective enrichment of target mycotoxin	Linear range: 0.05–200 μg/kg; LOD: 0.01–0.50 μg/kg; LOQ: 0.10–1.00 μg/kg; Recovery: 74.25–111.75 % (RSD: 2.08–9.01 %, *n* = 5)	[34]
PDA@Fe_3_O_4_-MWCNTs Magnetic SPE coupled with HPLC-FLD	AFB_1_, AFB_2_, AFG_1_, AFG_2_, OTA, OTB	Edible vegetable oils	Good water solubility and dispersibility; largely eliminating the influence of matrix effect	Linear range: 1–100 μg/L; LOD: 0.2–0.5 μg/kg; LOQ: 0.6–1.5 μg/kg; Recovery: 70.15–89.25 % (RSD: ≤ 6.4 %, *n* = 6)	[35]
rGO/AuNPs SPE coupled with UHPLC-MS/MS	AFB_1_, AFM_1_, OTA, ZEA, α-ZOL, β-ZOL, ZAN, α-ZAL, β-ZAL	Milk	Good adsorbability; adding AuNPs increases the distance between graphene layers and minimizes agglomeration	Linear range: 0.02–200 ng/mL; LOD: 0.01–0.07 ng/mL; LOQ: 0.02–0.18 ng/mL; Recovery: 70.1–111.1 % (RSD: 2.0–11.1 %, *n* = 5)	[36]
MIL-101(Cr)@Fe_3_O_4_ Magnetic SPE coupled with UHPLC-MS/MS	AFB_1_, AFB_2_, AFG_1_, AFG_2_, OTA, OTB, T-2, HT-2, DAS	Maize, wheat, watermelon, and melon	Magnetic separation and adsorption capabilities involving polar or nonpolar forces, hydrogen bonding forces, and π-π conjugation with mycotoxin-rich functional groups	Linear range: 0.2–100 ng/mL LOD: 0.02–0.06 μg/kg; LOQ: 0.08–0.2 μg/kg Recovery: 83.5–108.5 % (RSD: 1.6–10.4 %, *n* = 5)	[37]
Fe_3_O_4_@SiO_2_-NH_2_ Magnetic SPE coupled with ELISA	AFB_1_	Pixian douban	Rapid separation and enrichment under the external magnetic field; strong chemical stability, storage stability, and specificity combined with aptamer	Linear range: 0.5–2.0 ng/mL; LOD: 0.17 ng/mL; LOQ: 0.48 ng/mL; Recovery: 80.19–113.92 % (RSD: 2.30–7.28 %, *n* = 3)	[38]
HAS SPE coupled with HPLC-PHRED-FLD	AFB_1_	Vegetable oils	Outstanding adsorption properties due to the large number of functional group hydrogen bonding, hydrophobicity, and π-π interactions; minimizing the pretreatment time and the amounts of organic solvents	Linear range: 0.10–50 μg/kg; LOD: 0.03–0.09 μg/kg; LOQ: 0.1–0.3 μg/kg; Recovery: 66.9–118.4 % (RSD: ≤ 7.2 %, n = 6)	[39]
UIO-66-NH_2_@MIPs SPE coupled with HPLC	AFB_1_, AFB_2_, AFG_1_, AFG_2_	Wheat, rice, corn, soybean	Uniform and stable; the unique pore structure effectively improving the selective adsorption capacity; excellent affinity and selectivity	Linear range: 0.20–45 μg/kg; LOD: 0.06–0.13 μg/kg; LOQ: 0.24–0.45 μg/kg; Recovery: 74.3–98.6 % (RSD: 1.0–5.9 %, n = 6)	[40]
MWCNT-COOH + C_18_ SPE coupled with UPLC-MS/ MS	21 mycotoxins (AFs, OTA, OTB, ZEN, T-2, ZEN et al.)	Corn, wheat	Significantly reducing the matrix effect; high-throughput screening of various targets; greatly improving the detection efficiency	LOQ: 0.5–25 μg/L; Recovery: 75.6–110.3 % (RSD: 0.3–10.7 %, n = 5)	[41]
HNTs-HMIPs SPE coupled with HPLC-FD (Figure 1a)	ZEN	Rice corn, red beans, oats, wheat	Hollow imprinted polymer; excellent adsorption due to the loose and porous characteristics	LOD: 0.5 μg/kg; LOQ: 4.17 μg/kg (Oat), 1.8 μg/kg (Wheat); Recovery: 77.13–102.4 % (RSD: ≤ 5.59 %, n = 6)	[42]
**SPME**					
AuNPs SPME coupled with UHPLC-MS/MS	PAT	Apple juice, fresh apple, apple baby food, orange juice	Capillary monolithic column directly modified by AuNPs; high specificity and high affinity	Linear range: 8.11–8.11 × 10^3^ pmol/L; LOD: 2.17 pmol/L; Recovery: 85.4–106 % (RSD: 4.1–7.3 %, n = 5)	[43]
MAA-co-DVB SPME coupled with HPLC	AFB_1_, ZEN, STEH	Rice	High-strength micro/nanostructure containing a large number of acrylic groups forming hydrogen bonds with groups in the target structure; effectively overcoming the matrix effect	Linear range: 0.01–1.0 mg/kg; LOD: 0.689–2.030 μg/kg; LOQ: 5.36–14.4 μg/kg Recovery: 86.0–102.8 % (RSD: ≤ 4.8 %, n = 4)	[44]
Fe_3_O_4_@SiO_2_@Cu/Ni-NH_2_BDC Dispersive SPME coupled with HPLC-FLD	AFB_1_, AFB_2_, AFG_1_, AFG_2_	River water, well water, rice	Chemical bonds formed between three components making the adsorbent more stable and magnetic; rapid separation	Linear range: 0.11–79.2 ng/mL; LOD: 0.01–0.04 ng/mL; LOQ: 0.04–0.15 ng/mL; Recovery: 92.0–97.8 % (RSD: 4.1–7.6 %)	[45]
MOF+VB_3_ Dispersive SPME coupled with HPLC-FLD	PAT, OTA, AFB_1_, AFB_2_, AFG_1_, AFG_2_	Fruit juices, milk	Green organic linker; high surface area, high adsorption capacity, and excellent porosity to form a new green adsorbent	Linear range: 42.8–1 × 10^6^ ng/L; LOD: 11.3–48.2 ng/L; LOQ: 42.8–161.6 ng/L; Recovery: 64.0–87.0 % (RSD: ≤5 %, n = 3)	[46,47]

PDA-IL-NFsM, polydopamine and ionic liquid bifunctional nanofiber mat; HNTs-HMIPs, halloysite nanotubes hollow molecularly imprinted polymers; ST, sterigmatocystin; FB_1_, fumonisin B_1_; HT-2, HT-2 toxin; T-2, T-2 toxin; NIV, nivalenol; 3-AcDON, 3-acetylated deoxynivalenol; 15-AcDON, 15-acetylated deoxynivalenol; TEN, tentoxin; ALT, altenuene; ALS, altenusin; AME, alternariol monomethyl ether; AOH, alternariol; TEA, tenuazonic acid; ZEA, zearalenone; α-ZOL, α-zearalenol; β-ZOL, β-zearalenol; ZAN, zearalenone; α-ZAL, α-zearalanol; β-ZAL, β-zearalanol; OTB, ochratoxin B; DAS, ochratoxin B; PAT, patulin; STEH, sterigmatocystin; OTA, Ochratoxin A.

**Table 3 foods-12-03448-t003:** Application of various nanoscale materials for the rapid detection and screening of mycotoxins in food substrates.

Materials	Mycotoxins	Substrates	Action/Merits	Results	Ref.
Electrochemical Methods					
**MWCNTs/Fc-MOF**	AFB_1_	Walnut	Sensing substrate with large specific surface area for Ab incubation; stable internal reference electrochemical signals; improving the sensitivity and reliability by self-correction	Linear range: 10 fg/mL–100 ng/mL; LOD: 5.39 fg/mL; Recovery: 88.1–106 % (RSD: 1.2 %, n = 3)	[106]
**AuNPs/FeMOF@GO**	T-2	Bear	Larger specific surface area for aptamer loading; improving the electron transfer capacity	Linear range: 0.5–5.0 × 10^6^ pg/mL; LOD: 0.19 pg/mL; Recovery: 92.5–97.8 % (RSD: 3.2–5.5 %, n = 3)	[107]
**AuNPs/MnO_2_@GO**	T-2	Milk	Enhance the electrochemical active surface area for interface sensing and amplify the signal	Linear range: 2 fg/mL–20 ng/mL; LOD: 0.107 fg/mL; Recovery: 96.5–103.4 % (RSD: 3.8–4.3 %, n = 3)	[108]
**rGO/SnO_2_**	PTA	Apple juice	High surface area and excellent electrocatalytic performance	Linear range: 50–600 nM; LOD: 0.6635 nM; Recovery: 74.33–99.26 % (CV: 0.944–2.95 %, n = 3)	[109]
**PEG/AuNPs**	AFM_1_	Milk	Hydrated layer combining the hydrophilicity of PEG and high surface area of AuNPs; inhibiting protein corrosion and enhancing capacitance signal	Linear range: 20–300 pg/L; LOD: 7.14 pg/mL; Recovery: 101.6–105.5 % (RSD: <3%, n = 3)	[110]
**GO-CS/CeO_2_-CS**	AFM_1_	Milk	Strong conductivity, large specific surface area, and good redox performance; accelerating the electron transfer and amplifying the electrochemical signal	Linear range: 0.01–1 mg/L; LOD: 0.009 mg/L; Recovery: 96.15–104.25 % (RSD: 2.7–4.2 %, n = 3)	[111]
**ZnO NFs**	TEA	Tomato, orange	Highly specific surface area and excellent conductivity; a carrier for efficient immobilization of monoclonal Abs and Ab bioconjugates	Linear range: 5 × 10^−5^ –5 × 10^−1^ μg/mL; LOD: 1.14 × 10^−5^ μg/mL; Recovery: 95.71–120.3 % (RSD: 4.15–8.67 %, n = 3)	[112]
**AuNPs@Ce-TpBpy COF**	ZEN	Cornmeal	Adjustable pore size; high specific surface area; porous structure	Linear range: 1 pg/mL–10.0 ng/mL; LOD: 0.389 pg/mL; Recovery: 93.0–104.7 % (RSD: 1.26–5.54 %, n = 3)	[113]
**CuO@GO**	ZEN	Milk	Large electrochemical active surface area; high active electron transfer site and high conductivity	Linear range: 10–150 ng/mL; LOD: 0.012ng/mL; Recovery: 84.4–97.0 % (RSD: 1.01–1.34 %, n = 4)	[114]
**AuNPs/Ni-MOF**	AFB_1_	Rice flour	Larger specific surface area for improving electron transfer capacity and larger specific surface area, and amplifying electrochemical signals of the electrode	Linear range: 0.005–150.0 ng/mL; LOD: 0.001 ng/mL; Recovery: 98.7–101.3 % (RSD: 6.1–7.8 %, n = 3)	[115]
**Fluorescence Methods**					
**NMOFs**	AFB_1_	Maize	Low complexity; low interferences; high specificity and stability	Linear range: 0–3.33 ng/mL; LOD: 0.08 ng/mL; Recovery: 89.11–102.40 % (RSD: 3.17–5.39 %, n = 3)	[116]
**Zr-MOFs**	T-2	Milk, beer	Rich functional groups, easy to combine with auxiliary materials	Linear range: 0.5–100 ng/mL; LOD: 0.239 ng/mL; Recovery: 89.86–111.51 % (RSD: 2.0–2.9 %, n = 3)	[117]
**SWCNH**	AFB1	Soybean oil	Remarkable specificity and stability	Linear range: 10–100 ng/mL; LOD: 4.1 ng/mL; Recovery: 85.9–102.3 % (RSD: 3.59–5.46 %, n = 3)	[118]
**CuO NPs**	ZEN	Wheat, maize	High stability and biocompatibility; signal source and carrier for signal amplification; automatic sample pretreatment; high-throughput terminal detection	Linear range: 16.0–1600.0 μg/kg; LOD: 0.33 μg/kg; Recovery: 99.2–104.9 % (RSD: 0.7–5.1 %, n = 3)	[119]
**GO/Fe_3_O_4_**	AFB_1_, FB_1_	Peanut	Double fluorescence emission peaks quenching at the same time; effectively removing by magnetic separation to eliminate background interference	Linear range: 10 pg/mL–300 ng/mL; LOD: 6.7 pg/mL; Recovery: 92.0–97.0 % (RSD: 4.2–6.4 %, n = 3)	[120]
**FOQD**	PAT	Apple juice	Excellent fluorescence quenching ability; remarkable dispersibility in water	Linear range: 0.02–1 ng/mL; LOD: 0.01 ng/mL; Recovery: 95.0–103.0 % (RSD: 3.1–5.3 %, n = 3)	[121]
**QDNBs**	FB_1_, DON, ZEN	Wheat, maize	Good biocompatibility; the carrier for immobilization of Ab molecules	Linear range: 0.295–69.867 ng/mL; LOD: 0.87 ng/mL; Recovery: 78.61–122.31 % (CV: 1.10–14.36 %, n = 3)	[122]
**Al-MOFs**	AFB_1_	Tea	Good stability from respiratory effect; multiple forces for target toxin recognition	Linear range: 0.05–9.61 μM; LOD: 11.67 ppb; Recovery: 78.86–115.29 % (RSD: 0.83–7.72 %, n = 3)	[123]
**Zr-CAU-24**	AFB_1_	Walnut, almond beverages	Strong metal–ligand bond strength; high water stability; high sensitivity	Linear range: 0.075–25 μM; LOD: 19.97 ppb; Recovery: 91–108 %	[124]
**GO**	AFM_1_	Milk power	Protecting DNA aptamer from nuclease cleavage; target circulating signal amplification; high sensitivity	Linear range: 0.2–10 μg/kg; LOD: 0.05 μg/kg; Recovery: 98–126 % (SD: 1.48–6.3 μg/kg, n = 3)	[125]
**Colorimetric Methods**					
**AuNPs**	T-2	Wheat, maize	Acting as signal source; high sensitivity and portable	Linear range: 0.1–5000 ng/mL; LOD: 57.8 pg/mL; Recovery: 90.9–108.4 % (RSD: 0.7–7.21 %, n = 3)	[126]
**AuNBPs**	ZEN	Cornmeal	Highly sensitive signal source	Linear range: 0.02–0.80 ng/mL; LOD: 0.011 ng/mL; Recovery: 89.5–107.0 % (CV: 4.39–6.7 %, n = 3)	[127]
**MnO_2_**	OTA	Wheat flour, red wine	Excellent oxidase-like activity; using DNA regulate catalytic activity; high affinity for chromogenic substrates	Linear range: 0.05–33.35 ng/mL; LOD: 0.069 ng/mL; Recovery: 85.0–101.6 % (RSD: 1.33–6.83 %, n = 3)	[128]
**Au NPs@m-SiNPs**	AFs	Cornflakes, peanuts, butter, pecan nuts	Large surface area, high thermal stability, and chemical stability; acting as a fixed platform for AuNPs and direct immobilization of Abs	Linear range: 1–75 ng/mL; LOD: 0.16 ng/mL	[129]
**BP-Au**	DON	Maize, oat, millet	Remarkable color and photothermal conversion efficiency	Linear range: 0.1–8 ng/mL; LOD: 0.1 ng/mL; Recovery: 95.38–114.81 % (CV: 1.00–16.12 %, n = 3)	[130]
**ZnOBt**	AFs	Corn, almond	Color changes indirectly with targets through the oxidation–reduction reaction	Linear range: 0.5–20 ppb; LOD: 2.74 ppb; Recovery: 83.2–96.4 %; (RSD: 4.1–9.6 %, n = 5)	[131]
**Pt NPs/Fe-MOG**	FB_1_	Corn	Excellent peroxidase simulation activity and high affinity for substrates	Linear range: 0.01–2000.0 ng/mL; LOD: 2.7 pg/mL; Recovery: 98.7–101.9 % (RSD: 3.0–4.7 %, n = 3)	[132]
**FeO/GO, FeO@Au**	AFB_1_, OTA	Peanut	Large specific surface area; strong interactive affinity; high peroxidase activity; double-target detection	Linear range: 0.5–250 ng/mL; LOD: 0.15 ng/mL; Recovery: 87.3–102.5 % (RSD: 4.7–8.4 %, n = 3)	[133]
**Pt-CN**	AFB_1_	Peanut	Outstanding peroxidase simulation activity	Linear range: 1.0 pg/mL–10 ng/mL; LOD: 0.22 pg/mL; Recovery: 85.71–105.3 % (RSD: 2.23–8.33 %, n = 3)	[134]
**UiO-66-NH_2_**	ZEN	Wheat, maize	Ultrahigh loading capacity; excellent affinity with specific aptamer; excellent catalytic performance	Linear range: 0.01–100 ng/mL; LOD: 0.36 pg/mL; Recovery: 94.6–108.7 % (RSD: 1.0–8.7 %, n = 6)	[135]
**Other Strategies (LSPR and SERS)**					
**Aptamer-GNR** **LSPR**	OTA	Grape juice	Carrier for the aptamer specifically recognized the target; increasing local refractive index to lead to the shift of extinction peak	Linear range: 10 pM–100 nM; LOD: 12.0 pM; Recovery: 85.5–116.9 %	[136]
**PCPD-AgNPs** **LSPR**	AFB_1_	Peanut	Generating colorimetric signals based on LSPR changes of aggregated AgNPs for quantitative detection	Linear range: 0.2–6.0 ng/mL; LOD: 0.09 ng/mL; Recovery: 84.0–91.2 % (RSD: 0.6–1.8 %, n = 3)	[137]
**Au NBPs** **LSPR**	DON	Wheat, maize	Self-assembly into photonic crystals; enhancing the signal by coupling emission	Linear range: 0–2000 ng/mL; LOD: 57.93 ng/mL; Recovery: 93.7–107.5 % (RSD: 4.06–11.8 %, n = 4)	[138]
**AuNPS** **LSPR**	ZEN	Corn, wheat	Signal source; forming a stronger blue shift of LSPR peak, resulting in color change	Linear range: 0.04–2.96 ng/mL; LOD: 0.10 ng/mL (Naked eye), 0.07 ng/mL (Smartphone), 0.04 ng/mL (UV-spectra); Recovery: 76–112.6 % (RSD: 3.6–12.7 %, n = 3)	[139]
**AUNPs-3D SPCM** **SERS**	AFB_1_, OTA	Lily, Job’s tears seed, lotus seed	Great enhancement effect on SERS signal; good structural uniformity; accurate focusing position; overcoming the problem of SERS analysis and quantification	Linear range: 0.001–10 ng/mL; LOD: 0.034 pg/mL; Recovery: 80.23–116.20 % (CV: 6.3–7.16 %, n = 3)	[140]
**AgNPs@K30** **SERS**	AFB_1_, OTA, OTB	Rice	High roughness surface; anisotropic SERS substrate for capturing target toxin to produce the signal	Linear range: 0.5–500 µg/kg; LOD: 1.133 µg/kg; Recovery: 85.57–107.1 % (RSD: 5.84–8.71 %, n = 5)	[141]
**GO@Au-Au** **SERS**	FB_1_, AFB_1_, ZEN	Corn, peanut	Larger reaction interface; excellent stability and dispersibility; greatly improved the SERS activity and colorimetric signal	Linear range: 0.00046–10 ng/mL; LOD: 0.529 pg/mL; Recovery: 90.03–113.75 % (RSD: 2.79–13.48 %, n = 3)	[142]
**Au@SiO_2_** **SERS**	AFB_1_, OTA	Corn, rice, wheat	Improved plasmon resonance activity; chemical properties and biocompatibility; good dispersibility as SERS substrate	Linear range: 250 fg/mL–25 ng/mL; LOD: 0.24 pg/mL; Recovery: 87.0–108.0 % (RSD: 2.4–6.3 %, n = 4)	[143]

PEG, poly(ethylene glycol); CS, chitosan; ZnO NFs, flower-shaped nano-ZnO; SWCNH, single-walled carbon nanohorns; FOQD, fullerenol quantum dots; QDNBs, quantum dots nanobeads; AuNBPs, gold nanobipyramids; m-SiNPs, mesoporous silica nanoparticles; BP-Au, black phosphorus–gold; ZnOBt, zinc oxide bentonite nanocomposite; Pt NPs/Fe-MOG, platinum nanoparticles/Fe-based metal organic gel; Pt-CN, Pt supported on nitrogen-doped carbon amorphous; aptamer-GNR, aptamer-modified gold nanorods; PCPD-AgNPs, positively charged perylene diimide and AgNPs nanoprobes; SPCM, silica photonic crystal microsphere; CV, coefficient of variation; SD, standard deviation.

## Data Availability

Not applicable.

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
