# Peer review of "Nanoscale Materials Applying for the Detection of Mycotoxins in Foods"

_foods, 2023, doi:10.3390/foods12183448_

Round 1

Reviewer 1 Report

The presented material seems interesting and requires partial revision.

The authors need to make minor corrections to improve the article.

Comments are provided below.

1. Present the generalized scientific goal of the research in the abstract of the article

2. Present a generalized research scheme, graphic abstraction, for greater systematization of the review material

3. Increase the item “1. Introduction”, describe more fully the relevance and importance of the presented review study

4. Describe in more detail the financial costs of using different nanoscale materials in the SPE and SPME process for the detection of mycotoxins in foods

5. In conclusion, it is necessary to indicate possible options for the practical application of the results obtained, taking into account the financial costs of the proposed technology.

Author Response

Reviewer #1: The presented material seems interesting and requires partial revision.

The authors need to make minor corrections to improve the article.

Reply: Thank you for your review. The manuscript has been revised according to the comments of the reviewers and relative changes were marked. We hope that this manuscript can be accepted for Foods publication.

  1. Present the generalized scientific goal of the research in the abstract of the article

Reply: According to the comment, we have modified the abstract to clarify the goal for this review that have been marked in the revised manuscript. Thanks for the comment.

  1. Present a generalized research scheme, graphic abstraction, for greater systematization of the review material

Reply: According to the comment, we have supplied a graphic abstract in the revised manuscript. This graphic abstract makes the content of this review more systematically presented. Thanks for the comment.

  1. Increase the item “1. Introduction”, describe more fully the relevance and importance of the presented review study.

Reply: According to the comment, the Introduction part has been modified and added more content on the relevance and importance of this review (Line 53-65). Thanks for the comment.

  1. Describe in more detail the financial costs of using different nanoscale materials in the SPE and SPME process for the detection of mycotoxins in foods

Reply: Nanoscale materials for SPE and SPME are mostly used in the preparation of extraction and purification materials. During the detection process, the performance or dosage of the materials used are different for various toxin targets and actual samples, making it very difficult to calculate their costs. The focus of this review is to sort out the related applications of different nanoscale materials in the detection of mycotoxins by SPE and SPME, with less consideration of their financial costs. Thank you for the comment.

  1. In conclusion, it is necessary to indicate possible options for the practical application of the results obtained, taking into account the financial costs of the proposed technology.

Reply: In the conclusion, we summarize the corresponding relationship between the characteristics of different nanoscale materials and their specific applications (pretreatment materials, carriers of biorecognition molecules, signal sources, etc.), and further point out the direction of their research in the detection of mycotoxins. Please refer to the revised manuscript (Line 622-634, 639-641). Thank you for the comment.

Reviewer 2 Report

The article presents a revision about  recent studies on the  accurate and rapid detection of mycotoxins in food based on nanostructured materials,  with an emphasis on the role played by nanostructured materials with different properties in the detection process. This review provides valuable guidance for food safety monitoring and the development of advanced mycotoxin detection strategies. 

Author Response

Reviewer #2: The article presents a revision about recent studies on the accurate and rapid detection of mycotoxins in food based on nanostructured materials, with an emphasis on the role played by nanostructured materials with different properties in the detection process. This review provides valuable guidance for food safety monitoring and the development of advanced mycotoxin detection strategies.

Reply: Thank you for your kind comments on our work. It is hoped that this paper will provide a reference for the related research on nanoscale materials, especially its application in the detection of mycotoxins. The manuscript has now been revised based on comments from other reviewers. We hope it can be accepted for the publication of Foods.

Reviewer 3 Report

The review article is very interesting.

The authors present the recent research on nanostructured materials-based strategies for accurate or rapid detection of mycotoxins in food products, focusing on the properties and roles of various nanostructured materials in the detection process.

The topic is very relevant providing reference for the further development of nanostructured materials in the field of detection and guidance for the development of detection and control strategies for mycotoxins contamination in food matrices

The conclusions are consistent with the evidence and arguments presented and authors have accurately identified the limitations but also the perspectives of the study.

The references are appropriate, including some very relevant author’s previous experience in the field.

I suggest some corrections.

1 Tables 1 and 2 can not be accessed from the link provided. I suggest to be included in the main text.

Minor corrections

-        Line 90 Yuan et al., and all over the text, mention the references nr immediately after authors, not at the end of phrase or paragraph

-        Line 122 …rapid screening of polymycotoxins [mention the reference].

-        Line 212 Abs (and for some mycotoxins) give the full name when you use it for first time

-        Line 430 the energy of the donor can be transferred to the energy acceptor, correct as the energy of the donor can be transferred to the acceptor.

1.     In References section you may add the doi of the articles

Author Response

Reviewer #3: The review article is very interesting.

The authors present the recent research on nanostructured materials-based strategies for accurate or rapid detection of mycotoxins in food products, focusing on the properties and roles of various nanostructured materials in the detection process.

The topic is very relevant providing reference for the further development of nanostructured materials in the field of detection and guidance for the development of detection and control strategies for mycotoxins contamination in food matrices.

The conclusions are consistent with the evidence and arguments presented and authors have accurately identified the limitations but also the perspectives of the study.

The references are appropriate, including some very relevant author’s previous experience in the field.

I suggest some corrections.

Reply: Thank you for the comment. The comments and suggestions are very helpful for us to improve the quality of the manuscript. According to the comments, we have revised the manuscript and marked all the changes. We hope the enclosed manuscript can meet the requirements and be accepted for publication. Thanks again for your comment.

  1. Tables 1 and 2 cannot be accessed from the link provided. I suggest to be included in the main text.

Answer: According to the comment, we have supplied Table 1, 2, 3 in the revised manuscript. Thanks again for your review.

2.Minor corrections

(1) Line 90 Yuan et al., and all over the text, mention the references nr immediately after authors, not at the end of phrase or paragraph.

Answer: According to the comment, the manuscript has been modified. Thanks for the comment.

(2) Line 122 …rapid screening of polymycotoxins [mention the reference].

Answer: We are sorry for this mistake. The word “polymycotoxins” have been changed to “multi-mycotoxins” (Line 135).

(3) Line 212 Abs (and for some mycotoxins) give the full name when you use it for first time

Answer: Corresponding changes have been made in the revised manuscript (Line 141-142). We have carefully checked all abbreviations in the revised manuscript and ensured that the full name is provided on first use. Thanks for the comment.

(4) Line 430 the energy of the donor can be transferred to the energy acceptor, correct as the energy of the donor can be transferred to the acceptor.

Answer: Thank you for the comment. The language has been modified in the revised manuscript.

  1. In References section you may add the doi of the articles

Answer: Thank you for the comment. We have added the doi of the references in the revised manuscript, and ensured they are correct.

Reviewer 4 Report

1.        The year range of coverage of published reports reviewed in this article should be provided.

2.        A bibliometric analysis to emphasize the growth of this field through getting publication distribution over the years from Web of Science. To this effect, a bar diagram can be included by plotting the number of publications versus year.

3.        The introduction should include information on maximum permissible limits for mycotoxins recommended by various regulatory authorities should be included.

4.        After the introduction, an overview of toxicity of mycotoxins and conventional analytical methods for mycotoxins, their limitations and the need for rapid, sensitive, and on-site portable methods involving nanoscale materials.

5.        Section 3 is not divided properly into subtopics as it is summarized in Tables. For example, subtopics such as electrochemical, fluorescence, calorimetric, SERS etc.

6.        Altogether there are no figures and the authors should include at least 4 figures with a combination of pictures/plots/images of key/important/interesting findings from published reports.

7.        In Table 1, the recovery and reproducibility/precision data of the methods should be summarized. Also, the term ‘accuracy’ should be replaced with linear range or linear dynamic range, because ‘accuracy’ simply means ‘recovery’. But the recovery data in percentage is not given.

8.        Similarly, recovery and reproducibility/precision data should be provided in Table 2.

9.        All the nanomaterial notations in Tables 1 and 2 should be detailed in the footer of the respective tables.

 Minor editing of English language required

Author Response

Reviewer #4: 1. The year range of coverage of published reports reviewed in this article should be provided.

Answer: This review mainly summarizes the research results in related fields in recent years (2018-2023). Studies that are particularly informative are also considered, even if they were reported earlier. Corresponding additions and clarifications have been made in the revised manuscript (Line 14-23). Thanks for the comment.

  1. A bibliometric analysis to emphasize the growth of this field through getting publication distribution over the years from Web of Science. To this effect, a bar diagram can be included by plotting the number of publications versus year.

Answer: Thanks for the comment. This review summarizes the related research reported in recent years (2018-2023). According to the comment, we attempted to conduct a statistical analysis on reports related to the application of nanomaterials in the detection of mold toxins on Web of Science. However, regardless of various perspectives such as SPE, SPME, electrochemistry, or fluorescence, we were unable to effectively categorize these reports due to a significant amount of duplication in the literature. This is because nanoscale materials exhibit diverse functionalities in the detection process, leading to substantial overlap. Simply classifying them based on publication year does not accurately represent the actual research landscape within the scope of our study. Consequently, we are unable to provide a valid chart specific to the request. Thanks for your review.

  1. The introduction should include information on maximum permissible limits for mycotoxins recommended by various regulatory authorities should be included.

Answer: According to the comment, we have supplied the maximum permissible limits in the Introduction section (Table 1) Please refer to the revised manuscript for detailed changes. Thanks for your review.

  1. After the introduction, an overview of toxicity of mycotoxins and conventional analytical methods for mycotoxins, their limitations and the need for rapid, sensitive, and on-site portable methods involving nanoscale materials.

Answer: Thanks for the comment. This review has introduced the application of nanoscale materials for instrumental analysis of mycotoxin and for rapid detection and screening. The mentioned strategies of instrumental analysis are primarily geared towards accurate detection, whereas rapid detection has an advantage in large-scale screening of samples. According to the comment, we supplemented the required content in the revised manuscript, including the toxicity of mycotoxins (Table 1), the limitations of conventional detection methods (Line 73-80) and the necessity of detection strategies based on nanoscale materials (Line 307-314). Thanks for your review.

  1. Section 3 is not divided properly into subtopics as it is summarized in Tables. For example, subtopics such as electrochemical, fluorescence, calorimetric, SERS etc.

Answer: In Section 3, various applications of nanoscale materials in the field of rapid detection and screening of mycotoxins were discussed. These applications were categorized into sections “Carrier for biometric molecules”, “Signal source for rapid detection”, and “Other functions” based on their specific purposes. Table 2 (Now Table 3) provided an overview of these applications of nanoscale materials in various strategies. It should be noted that some nanoscale materials in research exhibited multiple functions, such as serving as both carriers of biorecognition molecules and sources of detection signals, making it challenging to differentiate their individual roles. Based on other comments, we have made improvements to Table 3.

  1. Altogether there are no figures, and the authors should include at least 4 figures with a combination of pictures/plots/images of key/important/interesting findings from published reports.

Answer: According to the comment, some representative six figures reported have been supplemented to the corresponding positions of the manuscript to support the relevant conclusions. Thank you for the comment.

  1. In Table 1, the recovery and reproducibility/precision data of the methods should be summarized. Also, the term ‘accuracy’ should be replaced with linear range or linear dynamic range, because ‘accuracy’ simply means ‘recovery’. But the recovery data in percentage is not given.

Answer: According to the comment, we re-adjusted the contents in Table 1 (Now Table 2) and supplied the “recovery” data. Please refer to the revised manuscript. Thank you for your review.

  1. Similarly, recovery and reproducibility/precision data should be provided in Table2.

Answer: Thank you for your review. Table 2 (Now Table 3) has illustrated the application of various nanoscale materials for rapid detection and screening of mycotoxins in food substrates. And the linear range and LOD were employed to demonstrate the merits of various strategies. According to the comment, we have supplied the recovery and reproducibility/precision data in the revised manuscript. Thanks for the comment.

  1. All the nanomaterial notations in Tables 1 and 2 should be detailed in the footer of the respective tables.

Answer: Thank you for the comment. We have added notes to the corresponding material in Tables 1 and 2 (Now Table 2 and 3) and checked the description of other abbreviations in the manuscript (Line 87-91, 320-324).

Round 2

Reviewer 4 Report

The authors have satisfactorily addressed all the comments raised by reviewers and therefore I recommend acceptance of this article for publication in Foods.

Minor editing of English language required